# The Effect of Omega-3 and Omega-6 Polyunsaturated Fatty Acids on the Production of Cyclooxygenase and Lipoxygenase Metabolites by Human Umbilical Vein Endothelial Cells

**DOI:** 10.3390/nu11050966

**Published:** 2019-04-27

**Authors:** Pedro Araujo, Ikram Belghit, Niels Aarsæther, Marit Espe, Eva Lucena, Elisabeth Holen

**Affiliations:** 1Institute of Marine Research, PO Box 1870 Nordnes, N-5817 Bergen, Norway; Ikram.Belghit@hi.no (I.B.); Marit.Espe@hi.no (M.E.); Elisabeth.Holen@hi.no (E.H.); 2Department of Biomedicine, University of Bergen, Jonas Lies vei 91, N-5009 Bergen, Norway; Niels.Aarsaether@uib.no; 3Department of Chemistry, Organic Analysis and Catalysis Laboratory, Simon Bolivar University, Caracas 1080A, Venezuela; evalucena@usb.ve

**Keywords:** polyunsaturated fatty acids, docosahexaenoic acid, eicosapentaenoic acid, α-linolenic acid, arachidonic acid, human umbilical vein endothelial cells, eicosanoids, prostacyclins, leukotrienes, resolvins

## Abstract

Although the correlation between polyunsaturated fatty acids (PUFA) and the production of pro- and anti-inflammatory metabolites is well documented, little is known about the simultaneous effect of different PUFA on the production of cyclooxygenase and lipoxygenase metabolites. The present research examines the association between different omega-3 (ω-3) and omega-6 (ω-6) PUFA and the release of four cyclooxygenase and six lipoxygenase metabolites in cell medium by human umbilical vein endothelial cells (HUVEC). The different combinations of ω-3 and ω-6 PUFA were prepared according to a full 2^4^ factorial design that enables studying not only the main effects but also the different interactions between fatty acids. In addition, interactions diagrams and principal component analysis were useful tools for interpreting higher order interactions. To the best of our knowledge, this is the first report addressing the combined effect of ω-3 and ω-6 PUFA on the signaling of prostaglandins, prostacyclins, leukotrienes and resolvins by HUVEC.

## 1. Introduction

Since the publication of the articles by Dyerberg and collaborators in the early seventies on the low incidence of heart disease in Greenland Eskimos [1,2], there has been a growing interest in the nutritional and pharmaceutical properties of polyunsaturated fatty acids (PUFA), especially those of the omega-3 (ω-3) family. Studies on the beneficial effects of ω-3 PUFA and the detrimental effects omega-6 (ω-6) PUFA on humans have been carried out extensively.

Long chain ω-3 PUFA including α-linolenic acid (18:3ω-3, ALA), eicosapentaenoic acid (20:5ω-3, EPA), docosapentaenoic acid (22:5ω-3, DPA) and docosahexaenoic acid (22:6ω-3, DHA) are a group of compounds possessing anti-inflammatory and immunomodulatory properties. They have demonstrated to be beneficial to cardiac [3,4], musculoskeletal [5], gastrointestinal [6] and immune systems in humans [7,8].

One of the most important functions of PUFA is related to their enzymatic conversion into eicosanoids. PUFA are released from membrane phospholipids by the action of various phospholipases and metabolized to different eicosanoids. Arachidonic acid (20:4ω-6, ARA) is the substrate for two classes of enzymes, cyclooxygenases (COX) which produce the 2-series of prostaglandins, prostacyclins and thromboxanes, and lipoxygenases (LOX) which catalyze the biosynthesis of hydroxyeicosatetraenoic acids (HETE) and the 4-series leukotrienes. EPA exhibits a similar metabolism to ARA, but it is metabolized to the 3-series prostaglandins, prostacyclins and thromboxanes through the action of the COX enzymes and 5-series leukotrienes and hydroxyeicosapentaenoic acids (HEPE) from LOX. DHA is mainly converted to D-series resolvins by LOX. The metabolism of PUFA is shown in Figure 1.

Studies on human cancer cells have indicated that ω-3 (e.g., EPA, DHA) and also ω-6 (e.g., ARA) PUFA are actively incorporated into endothelial cells and interconverted to anti- and pro-inflammatory mediators, such as eicosanoids, leukotrienes and resolvins, through the action of the COX (cyclic pathway) and LOX (linear pathway) enzymes [9,10]. These studies have also revealed that high levels of ω-6 PUFA in the Western diets is one of the main factors responsible for many maladies, including cardiovascular, cancer, autoimmune and inflammatory diseases [11,12,13,14,15,16,17], while ω-3 fatty acids exert suppressive effects [18,19].

The majority of reported studies are focused on evaluating particular fatty acids from specific families (either ω-3 or ω-6 PUFA). However, it is equally important to assess in which extent the interactions between different PUFA affect the production of COX and LOX metabolites. Unfortunately, such studies are still relatively scarce, partly due to their inherent multifactorial character and their susceptibility to confounding factors that in turn might unveil correlations between variables when there is not actually a correlation [20]. For instance, nutritional studies on the effect of ω-3 PUFA generally use marine oils as a source of EPA and DHA to induce significant beneficial variations on different experimental groups. These variations might be the result of confounding factors, considering for example that marine oils are also rich in vitamin D.

Confounding factors may contribute to inconsistent or conflicting results as reported in studies on the production of inflammatory mediators by salmon and cod head kidney cells supplemented with ω-3 PUFA [21,22] or in human plasma from patients exposed to an anti-inflammatory treatment of salmon and vitamin D [23] where it was observed that ω-3 PUFA increases the production of inflammatory eicosanoids rather than decreasing it. This puzzling finding was rationalized as the incorporation of exogenous ω-3 PUFA at the expense of releasing endogenous ω-6 PUFA from the cell membrane which in turn is converted into inflammatory eicosanoids. This example highlights the significance and complexity of confounding variables and the importance of scrutinizing the conclusions derived from nutritional studies, especially those claiming that a specific diet component will cause or prevent certain diseases.

Human umbilical vein endothelial cells (HUVEC) can be regarded as a valuable tool for studying different biological aspects of endothelial cells. Their availability, easy to isolate in a pure form, low cost, rapid culturing and proliferation in simple laboratory settings (e.g., medium containing fetal bovine serum) made them a very attractive system for modelling the influence of ω-3 and ω-6 PUFA on the production of their associated pro- and anti-inflammatory biomarkers through the cyclic (COX enzyme) and linear (LOX enzyme) pathways.

Published studies on the production of eicosanoids in cell culture systems supplemented with ω-3 and ω-6 PUFA are generally focused on the effect of individual PUFA in a univariate way without considering the potential effect of their interactions on eicosanoid production [24,25].

The present research aims at studying the production of cyclooxygenase and lipoxygenase metabolites by HUVEC exposed to different combinations of PUFA. The research assesses specifically the levels of prostaglandins (PGE_2_, PGE_3_), prostacyclins (6-keto-PGF_1α_, Δ17-6-keto-PGF_1α_), resolvins (RvD_1_, RvD_2_, RvD_3_, RvD_4_ and 17-epi-RvD_1_) and leukotriene (LTB_4_) released into endothelial cell growth basal medium-2 (EBM-2) by HUVEC exposed to ARA, EPA, DHA and ALA, 18:3n-3 by means of a full factorial design and a previously developed solid phase extraction and liquid chromatography tandem mass spectrometry (SPE-LC-MS/MS) method [26]. Through this research, it was observed the inhibitory effect of DHA and ALA on the production of pro-inflammatory COX metabolites, the lack of correlation between EPA and production of anti-inflammatory prostaglandin or prostacyclin, the association between ARA and increased production of all pro-inflammatory metabolites, and the enhanced production of LOX metabolites by DHA. To our knowledge, the present research is the first report addressing the influence of single and combined ω-3 and ω-6 PUFA on the simultaneous production of eicosanoids and resolvins’ metabolites by HUVEC.

## 2. Materials and Methods

### 2.1. Reagents

Prostaglandins: PGE_2_ (99%), PGE_3_ (98%); prostacyclins: 6-keto-PGF_1α_ (98%), Δ17-6-keto-PGF_1α_ (98%); leukotriene: LTB_4_, (97%); resolvins: RvD_1_ (95%), RvD_2_ (95%), RvD_3_ (95%), RvD_4_ (95%), RvD_5_ (95%) and deuterated internal standards: PGE_2_-d_4_ (99%), 6-keto-PGF_1α_-d_4_ (99%), LTB_4_-d_4_ (97%) and RvD_2_-d_5_ (95%) were purchased from Cayman Chemical (Ann Arbor, MI, USA).

Acetonitrile (99.8%), ethanol (99.8%) acetic acid (99%) and formic acid (98%) were purchased from Sigma-Aldrich (St. Louis, MO, USA), 2-propanol (HPLC grade, 99.9%) from Merck (Darmstadt, Germany), a Millipore Milli-Q system was used to produce ultra-pure water 18 MΩ (Millipore, Milford, MA, USA). EPA (99%), ARA (95%), DHA (98%) and ALA (99%) were from Sigma-Aldrich (Oslo, Norway). Complete EBM-2 contained EBM^TM^-2 basal medium supplemented with 0.1% heparin, 0.1% R^3^-IGF−1 solution, 0.1% ascorbic acid, 0.04% hydrocortisone, 0.4% h-FGF-B, 0.1% h-EGF, 0.1% GA−1000 and 2% fetal bovine serum (FBS, cat# 14-801F) was from BioWhittaker (Petit Rechain, Belgium).

### 2.2. Cell Culture

The HUVEC were bought from Sigma-Aldrich and incubated in culture plates with complete EBM-2 medium supplemented with 0.1% heparin, 0.1% R^3^-IGF−1 solution, 0.1% ascorbic acid, 0.04% hydrocortisone, 0.4% h-FGF-B, 0.1% h-EGF, 0.1% GA−1000 and 2% FBS. Fifteen PUFA ethanolic solutions, suggested by the factorial design (Table 1), were prepared at a level of concentration of 6.1 M of each fatty acid. Aliquots of 15 µL were taken from every solution by attaching the fatty acids to FBS at a ratio of 2:1 for 18 h at room temperature and diluting with EMB-2 medium to a final a concentration of 46 μM of fatty acid. A blank solution was made by adding FBS and ethanol (the solvent used to dissolve the fatty acids) and diluting with EMB-2 medium. Once the number of cells reached 10^5^ cells/well, the medium was removed and the EMB-2 solutions containing EPA, ARA, DHA and ALA were added according to a 2^4^-full factorial design (Table 1). Cultures were prepared in triplicate (16 × 3) and incubated at 37 °C in humidified atmosphere of 95% air and 5% CO_2_. After 24 h, the supernatants were collected and stored at –80 °C until extraction.

### 2.3. Extraction and Quantitative Measuring of Cyclooxygenase and Lipoxygenase Metabolites

The SPE protocol and LC-MS/MS quantification were based on a previous published method [26]. Briefly, an aliquot of 100 μL of a mixture of internal standards (180 ng/mL PGE_2_-d_4_, 45 ng/mL 6-keto-PGF_1α_-d_4_, 40 ng/mL RvD_2_-d_5_ and 30 ng/mL LTB_4_-d_4_) was added to 1 mL of sample. Ethanol (175 μL) and acetic acid (20 μL) were added and the mixture was vortex-mixed and applied to a SPE column (Agilent, ASPEC Bond Elute C18, 500 mg, 3 mL, Santa Clara, CA, USA) which had been preconditioned with 2 mL of methanol and 2 mL of water. The cartridge was washed with 4 mL of distilled water and 4 mL of hexane to remove peptides and salts as well as polar and nonpolar interfering substance. The analytes were eluted with 1 mL of hexane/ethyl acetate (1:2 *v*/*v*), collected into a glass tube and the solvent was evaporated under a stream of nitrogen gas. The dried sample was redissolved in 70 μL of acetonitrile, vortex-mixed 30 s, centrifuged at 1620 g for 3 min and transferred to an auto sampler vial for LC-MS/MS quantitative analysis by using electro spray ionization in negative mode. The quantification was carried out by using the internal standard calibration technique described elsewhere [26] by dissolving PGE_2_, PGE_3_, 6-keto-PGF_1α_, Δ17-6-keto-PGF_1α_, LTB_4_, RvD_1_, 17-epi-RvD_1_, RvD_2_, RvD_3_ and RvD_4_ in complete EBM-2 medium and adding the internal standards at the above indicated concentration levels. The analysis of the extracted ion chromatograms of the analytical species revealed that the characteristic mass fragmentation patterns were clearly distinguished from each other, indicating that the analysis was selective towards deuterated and non-deuterated analytes.

### 2.4. Statistics

To evaluate the main effects and their interactions, multifactor ANOVA was performed to determine whether or not there are significant differences between the means of the analysed metabolites with or without the fatty acids and the significance established by means of F-ratios and expressed as *p*-values by using Statgraphics Centurion XVI (Version 16.1.11, StatPoint Technologies, Inc., Warrenton, VA, USA). Tukey’s test was employed as correction procedure for multiple testing. The interactions between two factors were analysed by means of interaction diagrams described elsewhere [27,28].

The results were submitted to principal component analysis (PCA) in order to detect meaningful relationships between the high order interactions and highlight differences and similarities in the data set by using Statistica™ (data analysis software system) version 13.4.0.14, TIBCO Software Inc. (2018).

## 3. Results

The released PGE_2_, PGE_3_, 6-keto-PGF_1α_, Δ17-6-keto-PGF_1α_, LTB_4_, RvD_1_, 17-epi-RvD_1_, RvD_2_, RvD_3_ and RvD_4_ into EBM-2 by HUVEC after exposure to single or combined DHA, EPA, ALA and ARA and by using the described 2^4^-full factorial design are shown in Table 1 as average values of three independent replicates (individual data are presented in Appendix A). The term CODE in Table 1 is used to summarize the various conditions dictated by the design. For example, ALL as a code indicates the presence of DHA, EPA, ALA and ARA (condition +1, +1, +1, +1, respectively), while ALL-DHA indicates the absence of DHA and the presence of EPA, ALA and ARA (condition −1, +1, +1, +1, respectively).

### 3.1. Main Effects Evaluation

The effect of the main terms were analysed by computing the average values of every metabolite without (−1) and with (+1) PUFA (Table 2) and their significance expressed as *p*-values (Figure 2, Figure 3 and Figure 4 and Appendix A).

The presence of DHA (+1 level) caused a statistically significant decrease in the production of pro- and anti-inflammatory prostaglandins and prostacyclins (PGE_2_, 6-keto-PGF_1α_, Δ17-6-keto-PGF_1α_ and PGE_3_) and a significant increase in the production of pro- and anti-inflammatory metabolites from the LOX pathway (leukotriene and resolvins) (Table 2, Figure 2, Figure 3 and Figure 4 and Appendix A). The highest and lowest decrease in production were exhibited by PGE_2_ (73%) and PGE_3_ (33%) respectively, while RvD_4_ (1673%), 17epi-RvD_1_ (1522%) and RvD_1_ (1278%) showed remarkable increases in production.

The addition of EPA did not affect the production of the prostaglandins and prostacyclins (Figure 1) but LTB_4_ that was increased in 60% (Figure 3, Table 2). The results after exposing HUVEC to EPA (Table 2, Figure 3 and Figure 4) are characterized by significant increases in LTB_4_ (60%; *p* < 0.000) and RvD_2_ (30%; *p* = 0.002) and significant decreases (*p* < 0.000) in 17epi-RvD_1_ (29%) and RvD_4_ (21%). The rest of the resolvins (RvD_1_ and RvD_3_) production did not change after exposure to EPA.

Addition of ALA to HUVEC influenced the production of prostaglandins, prostacyclins, leukotriene and RvD_2_, RvD_3_ and RvD_4_ (Figure 2, Figure 3 and Figure 4). The only metabolites that showed significant increase in production were 6-keto-PGF_1α_ (30%; *p* < 0.000), LTB_4_ (83%; *p* < 0.000) and RvD_4_ (13%; *p* = 0.021) as indicated in Table 2 and Appendix A.

The exposure to ARA increased significantly the production of all pro- and anti-inflammatory prostaglandins, prostacyclins, LTB_4_ and all the resolvins (*p* < 0.000) with the exception of RvD_3_ (Figure 4) that remained unaffected after exposure to ARA (*p* = 0.427). Table 2 shows that, after exposure to ARA, the highest increase in metabolites from the COX pathway was exhibited by PGE_2_ (3241%) followed by PGE_3_ (537%), 17epi-RvD_1_ (461%) and 6-keto-PGF_1α_ (228%), while LTB_4_ (188%) showed the highest increase for metabolites from the LOX pathway followed by 17epi-RvD_1_ (103%).

### 3.2. Interaction Effects Evaluation

Interaction diagrams (Figure 5 and Figure 6) were used to explain the effect of the different two-term interactions on the production of COX- and LOX-derived metabolites. The diagrams were constructed by allocating every pair of PUFA on a two-dimensional graph with mutually perpendicular axes. The graph was divided into four quadrants and the values in each quadrant were computed according to the levels of the 2^4^ factorial design as explained in Appendix A. For example, when studying the interaction EPA and DHA, they are designated as *x*-axis and *y*-axis respectively. The intersecting *x*- and *y*-axes divide the coordinate plane into four quadrants (Appendix A). The quadrant I (levels +1,+1) is the average value of all the experiments containing both ω-3 PUFA (e.g., ALL, ALL-ARA, ALL-ALA, DHA+EPA in Table 1, Appendix A), the quadrant II (levels −1, +1) is the average value of all the experiments without EPA but containing DHA (e.g., ALL-EPA, DHA+ALA, DHA+ARA, DHA in Table 1, Appendix A), the quadrant III (levels −1,−1) is the average value of all the experiments without EPA or DHA (e.g., ALA+ARA, ALA, ARA, EtOH in Table 1, Appendix A) and the quadrant IV (levels +1, −1) is the average value of all the experiments containing EPA but not DHA (e.g., ALL-DHA, EPA+ALA, EPA+ARA, EPA in Table 1, Appendix A). The statistical significance of the interactions are showed in Figure 2, Figure 3 and Figure 4 and Appendix A.

The interaction effect of DHA×EPA on reducing the production of inflammatory and anti-inflammatory prostaglandins and prostacyclins by HUVEC was not significant (*p* > 0.05) as indicated in Figure 2. Figure 5 shows that the initial production of PGE_2_ without these PUFA (III quadrant: 153.9 ± 3.4 pg) is dramatically reduced after adding DHA (II quadrant: 37.9 ± 3.4 pg), but it is marginally reduced after adding EPA (IV quadrant: 149.2 ± 3.4 pg). In addition, the reduction after adding DHA and EPA (I quadrant: 44.53 ± 3.6 pg) is statistically equivalent to the effect of adding DHA alone, hence the significance of the interaction DHA×EPA (*p* = 0.105) is disregarded. A similar behavior was observed for the rest of the prostaglandins and prostacyclins when the interaction DHA×EPA is considered (Figure 5).

The results in Figure 6 for the production of LTB_4_ without EPA or DHA (25.5 ± 2.5 pg) revealed a considerable enhancement when the HUVEC were exposed either to DHA (II quadrant: 75.6 ± 2.5 pg) or EPA (IV quadrant 46.5 ± 2.5 pg). However, their combined effect DHA×EPA results in a much higher and significant release (I quadrant: 115.6 ± 2.6 pg).

The interaction term DHA×EPA revealed to be significant (Appendix A) in decreasing the production of 17epi-RvD_1_ (*p* = 0.000, Figure 3), RvD_3_ (*p* = 0.006, Figure 4) and RvD_4_ (*p* = 0.013, Figure 4) and increasing the production of RvD_2_ (I quadrant: 5.7 pg, Figure 6) compared to DHA alone (II quadrant: 4.3 pg, Figure 6), which represent an increasing of 32.6% (Figure 3, *p* = 0.002). Although the effect of the interaction DHA×EPA was not significant for RvD_1_ (*p* = 0.075), a slight tendency to decrease (I quadrant: 19.5 pg) was observed compared to DHA alone (II quadrant: 20.6 pg).

The interaction DHA×ALA contributed in decreasing the production of all inflammatory or anti-inflammatory prostaglandins and prostacyclins (Figure 2, Appendix A) compared to the system without these PUFA. In contrast, the interaction DHA×ARA significantly promoted the production of all COX-derived metabolites (Figure 2). The interaction DHA×ARA played a significant role in increasing the production of RvD_1_ (*p* < 0.000) but not DHA×ALA (*p* = 0.130) (Figure 3). Both interactions, DHA×ALA and DHA×ARA, significantly enhanced the production of LTB_4_, 17epi-RvD_1_, RvD_2_, RvD_3_ and RvD_4_ (Figure 3 and Figure 4).

The interaction EPA×ALA decreased the production of PGE_2_ and Δ17-6-keto-PGF_1α_ (*p* < 0.000), while the production of 6-keto-PGF_1α_ (*p* < 0.000) is increased (Figure 2). This particular interaction decreased the production of all anti-inflammatory resolvins and PGE_3_ (*p* = 0.384). However, the observed reduction was statistically significant only for RvD_1_ (*p* = 0.002) and RvD_2_ (*p* < 0.000) (Figure 3).

The interaction EPA×ARA did not affect the production of any of the analyzed pro-inflammatory but the anti-inflammatory mediators (except for RvD_1_). While the interaction ALA×ARA was not significant for 6-keto-PGF_1α_ (*p* = 0.969), RvD_1_ (*p* = 0.503) and RvD_3_ (*p* = 0.211).

The most salient feature of the triple and quadruple interaction terms is that EPA×ALA×ARA and DHA×EPA×ALA×ARA influence all the pro- and anti-inflammatory metabolites, except for PGE_3_ (*p* = 0.562) in the former and PGE_2_ (*p* = 0.523) and PGE_3_ (*p* = 0.460) in the latter.

### 3.3. Principal Component Analysis (PCA)

The data set (Appendix A) was arranged in a *m* × *n* matrix, where *m* represents the experimental conditions and *n* represents the different ten measured COX- and LOX-metabolites. The 50 × 10 matrix was standardized by subtracting their means and dividing by their standard deviations. The PCA score and loading plots, describing relationships between PUFA and between metabolites, respectively were computed and shown in Figure 7. The score plot in Figure 7a revealed that the information retained by PC1 and PC2 (explaining 43.38 and 34.74% of the total data variability) is mainly associated with DHA and ARA, respectively. These PUFA allow discriminating four experimental patterns according to their presence (+DHA or +ARA) or absence (-DHA or -ARA). A high degree of overlapping was also observed between the various experimental conditions and biological replicates, including the two types of controls (-EtOH and +EtOH) which might indicate a similar mechanism in the production of the measured biomarkers. The loading plot (Figure 7b) revealed that the inverse correlation between LOX metabolites (negative PC1 loading values) and COX metabolites (positive PC1 loading values) is responsible for the observed discrimination patterns. The contribution of LTB_4_ (−0.462) in the discrimination process was lower than the resolvins, while the contribution of the anti-inflammatory PGE_3_ (0.398) was lower than the inflammatory prostaglandins and prostacyclins. The highest absolute PC1 loading values were exhibited by RvD_1_ (−0.896) and PGE_2_ (0.622) and consequently their contribution to the discrimination process was higher than the rest of the metabolites.

## 4. Discussion

The multitude of potential mechanisms and the complexity of the PUFA has made it difficult to fully understand the actions of these fatty acids within inflammatory processes. Thus, in the present study, a full factorial design was used to study the release of pro- and anti-inflammatory lipid mediators into EBM-2 by HUVEC exposed to both individual and combined PUFA, more specifically DHA, EPA, ALA and ARA.

### 4.1. Main Effects

#### 4.1.1. DHA Effect

The findings (Figure 2, Figure 3 and Figure 4 and Table 2) confirmed the strong relationship between addition of exogenous DHA and increased levels of all the metabolites from the LOX pathway (especially the resolvins) and decreased levels of all the metabolites from the COX pathway (prostaglandins and prostacyclins). The observed inhibition of prostaglandins and prostacyclins by DHA is in agreement with data from human umbilical cord [29] and bovine aortic [30,31,32,33,34] endothelial cells, respectively.

#### 4.1.2. EPA Effect

Interestingly, we found no significant association between EPA and increased production of anti-inflammatory PGE_3_ or Δ17-6-keto-PGF_1α_. The levels of PGE_3_ and PGE_2_ remained unaltered after adding EPA (Table 2) that is in stark contrast to the reported competition process between ω-3 PUFA and ARA for binding to the cyclooxygenase active site and thereby inhibit formation of the 2-series prostaglandins that are derived from ARA [35]. For instance, some studies have observed that an increased intake of EPA suppressed the synthesis of ARA-derived eicosanoids and increased the formation of the analogous EPA derived mediators [7,36,37,38,39,40]. The incorporation of EPA and release of ARA has already been reported [41]. However, it is important to point out that this effect occurs in a dose-response and time-dependent fashion [41,42]. The findings in the present research are similar to those reported in previous HUVEC experiments [43], where the lower affinity of COX enzymes for ω-3 EPA was ascribed to a low rate of EPA oxygenation by COX−1 as explained elsewhere [35]. EPA and ARA bind to the cyclooxygenase active site by positioning their C−13 below Tyr-385 for hydrogen abstraction. The presence of an additional double bond in EPA decreases its flexibility in the cyclooxygenase active site resulting in a strained binding orientation and a misalignment between C−13 and Tyr-385, which is presumably responsible for the low rate of EPA oxygenation and consequently a low conversion into PGE_3_ [35]. The previous observations are confirmed in Figure 5, where the simultaneous effect of EPA×ARA revealed a constant production of PGE_2_ regardless of the presence of EPA (187.0 ± 3.6 pg without EPA versus 187.2 ± 3.4 pg with EPA) but a remarkable production of PGE_2_ by the presence of ARA (4.8 ± 3.4 pg versus 187.0 ± 3.6 pg). Further investigation is necessary to assess the influence of the concentration of EPA in the production of PGE_3_. In addition, the production of anti-inflammatory metabolites from EPA through the LOX pathway such as LTB_5_ or the E-series resolvins should be investigated to verify whether the production of these metabolites are preferred over PGE_3_.

#### 4.1.3. ALA Effect

In general, the effect of ALA brought about a considerable reduction in the production of anti- and pro-inflammatory metabolites from the COX pathway (Figure 2 and Figure 3, Table 2 and Appendix A), indicating a possible inhibition of the COX-enzymes by the action of ALA which in turn might cause the observed remarkable enhancement of the LOX pathway as reflected in the 83% increase in LTB_4_. The inhibitory role of ALA has been reported in a study aiming at isolating compounds with COX inhibitory activity [44].

#### 4.1.4. ARA Effect

The levels of all pro-inflammatory metabolites were remarkably high in all the preparations containing ARA (Figure 2, Table 2). These results were in accordance with the well-known fact that pro-inflammatory prostaglandins, prostacyclins and leukotriene are promoted via COX and LOX pathways from ARA. In addition, the observed increase in both EPA (PGE_3_ and Δ17-6-keto-PGF_1α_) and DHA (with the exception of RvD_3_) derived metabolites (Figure 2, Figure 3 and Figure 4, Table 2) could indicate that, during the incubation period of 24 h, ARA is incorporated in the cell membrane at the expense of EPA and DHA. The incorporation of ARA into the cell membrane has been already reported by using radioactive free ARA [45]. Furthermore, some studies have observed increased levels of LTB_5_ (EPA metabolite) in skin cells from dogs fed n-6/n-3 diet ratios of 5/1 and 10/1 [46]. The amounts and types of synthesized eicosanoids are determined by several factors, among them the availability of released PUFA, the activity of the cyclo- or lipoxygenase and the cell type [47,48].

### 4.2. Two-Term Interactions

#### 4.2.1. DHA×EPA, DHA×ALA, DHA×ARA

The interaction DHA×EPA showed a general non-significant decrease in those metabolites derived from the COX pathway and a statistically significant increase in all the metabolite derived from the LOX pathway (leukotriene and resolvins) with the exception of RvD_1_ that was not significant at the 95% but at the 99% confidence level. This result might indicate a synergist effect between the stimulatory effect of DHA on LOX expression and inhibitory effect on COX expression. The interactions DHA×ALA and DHA×ARA were statistically significant for all the COX-derived metabolites. The behavior of the former interaction was characterized by a decreased production of all the analyzed COX-derived metabolites, while the latter exhibited a consistent increment in production of the equivalent metabolites. The observed behavior is in accordance with the reported inhibitory effect of DHA and ALA [29,44] and widely reported stimulatory effect of ARA on COX-2 expression. Both interactions, DHA×ALA and DHA×ARA, showed a positive impact on the production of resolvins as a result of DHA stimulation.

#### 4.2.2. EPA×ALA, EPA×ARA

The interaction EPA×ALA had a negative impact on PGE_2_ and the two EPA derived metabolites (Δ17-6-keto-PGF_1α_ and PGE_3_). The production of PGE_2_, Δ17-6-keto-PGF_1α_ and PGE_3_ is governed by the substrates ARA and EPA and the COX enzymes. However, as explained above, EPA exhibited a low oxygenation by COX−1, which could bring about a low production of Δ17-6-keto-PGF_1α_ and PGE_3_ [35]. In addition, some studies have demonstrated that ALA has a strong selectivity towards COX-2, which in turn has a negative impact on the biosynthesis of PGE_2_ [38]. Although LTB_4_ and all the analyzed resolvins from the D-series are LOX-5 catalyzed derivatives, the production rate of LTB_4_ by the action of the interactions EPA×ALA and EPA×ARA was higher than those observed for resolvins. This result might be attributed to the lack of exogenous DHA, the release of ARA from the cell membrane after EPA incorporation [43] and the ALA COX-2 inhibitory effect [44] that promotes the signaling of ARA through the LOX pathway.

#### 4.2.3. ALA×ARA

In general, the interaction ALA×ARA brought about an increase in all the studied metabolites. The positive effect of this interaction on the production of Δ17-6-keto-PGF_1α_ and PGE_3_ is notable despite the lack of exogenous EPA. Two possible mechanisms of action could explain the observed effect. Firstly, the incorporation of ARA and further release of EPA and secondly, the biosynthesis of EPA from ALA. It has been reported that ALA is a substrate for the synthesis of EPA, but the conversion rates are low [49,50]. In light of the present research, it is possible to hypothesize that exogenous ALA might contribute to enhancing the EPA production and consequently its associated COX metabolites.

### 4.3. Three- and Four-Term Interactions

The triple and four term interactions containing the term ALA×ARA exhibited the highest numbers of significant effects compared to those without this particular interaction term (Figure 2, Figure 3 and Figure 4, Appendix A). For example, EPA×ALA×ARA was not statistically significant for PGE_3_ but significant for the rest of the metabolites, DHA×ALA×ARA and DHA×EPA×ALA×ARA were not significant for RvD_1_, RvD_3_ and PGE_2_, PGE_3_, respectively (both with eight significant effects). The interactions DHA×EPA×ALA and DHA×EPA×ARA were not significant for PGE_2_, 6-keto-PGF_1α_, PGE_3_, RvD_4_ and PGE_2_, Δ17-6-keto-PGF_1α_, PGE_3_ and RvD_1_ (both with six significant effects). It is remarkable that PGE_3_ was not significant in all three- and four-PUFA interactions containing exogenous EPA, indicating probably its inhibitory effect on COX enzyme or the conversion of EPA to the prostanoid PGH_3_, which can be isomerized to thromboxane A3 (TXA_3_) and PGD_3_ (Figure 1). The interpretation of the three- and four-PUFA interactions is considerably more complicated, especially bearing in mind that the behavior of ω-3 and ω-6 PUFA and their derivatives in biological systems have been regarded as more intricate than previously recognized [51]. The PCA was proposed to facilitate the interpretation process and detect important relationships between the interactions. The PCA plots confirmed that single ARA or that combined with EPA or ALA (e.g., ARA, EPA×ARA, ALA×ARA and EPA×ALA×ARA in the quadrant −DHA/+ARA of Figure 7a) are consistently associated with high levels of COX metabolites and low levels of resolvins (Figure 7b) and that the effect of the interaction EPA×ARA is equivalent to the effect of ARA alone (Figure 7a). Conversely, replacing ARA by DHA to evaluate the impact of the counterpart preparations DHA, DHA×EPA, DHA×ALA and DHA×EPA×ALA (quadrant +DHA/–ARA of Figure 7a) brings about a significant reduction in the levels of COX metabolites and an increase in the levels of LOX metabolites. The lowest levels of COX and LOX metabolites were exhibited by preparations where DHA and ARA were not added (EPA, ALA, EPA×ALA and control with ethanol in the quadrant –DHA/–ARA of Figure 7a) indicating probably a low affinity of the COX and LOX enzymes towards EPA and a low rate of interconversion of ALA into EPA. The highest levels of LOX metabolites were found in those preparations containing both DHA and ARA (DHA×ARA, DHA×EPA×ARA, DHA×ALA×ARA and DHA×EPA×ALA×ARA in the quadrant +DHA/+ARA of Figure 7a). The levels of LTB_4_ were consistently higher than the control in all the PUFA preparations.

## 5. Conclusions

The role of lipid mediators in human health, signaling and regulation processes is partly determined by the specific PUFA from which the pro- or anti-inflammatory metabolites are derived. Thus, the present research has examined the association between different ω-3 and ω-6 PUFA and the release of ten different COX and LOX metabolites in cell culture medium by HUVEC. The present research confirmed earlier published results on the anti-inflammatory role of ω-3 PUFA (DHA and ALA), and the lack of effectivity of EPA towards the production of PGE_3_ in a HUVEC model. However, the non-significant production of PGE_3_ by EPA seems to be controversial and it has been reported previously in mammalian cell models, including HUVEC. In addition, exposing the cells to exogenous ARA resulted in a high production of COX metabolites. The high levels of COX metabolites were suppressed when ARA was combined with DHA, ALA or DHA×ALA. DHA enhanced all the LOX metabolites (LTB_4_ and resolvins), while ALA enhanced only the levels of the LTB_4_ but had no effect on resolvins. The present findings have indicated that DHA but not EPA is the driving force behind the beneficial effects of these ω-3 PUFA and that the actions of ω-3 and ω-6 PUFA and their derivatives on inflammatory processes involve more complex mechanisms than previously recognized. Consequently, future investigations are necessary to access the effect of PUFA, especially EPA, at different concentrations and in a time-dependent fashion.

## Figures and Tables

**Figure 1 nutrients-11-00966-f001:**
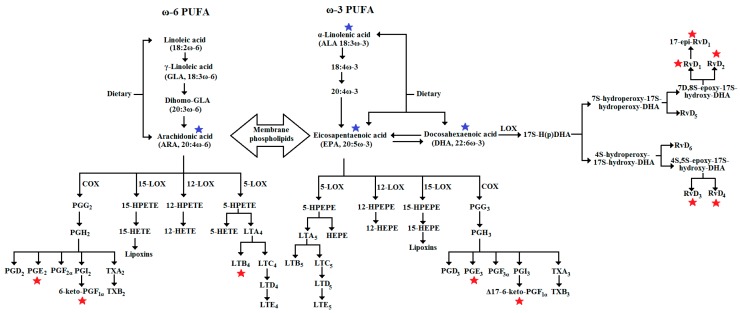
Metabolism of different ω-3 and ω-6 PUFA. EPA and ARA compete for the same cyclooxygenase (COX) and lipoxygenase (LOX) enzymes and they are converted into prostaglandins, prostacyclins and thromboxanes by COX enzymes and into leukotrienes, lipoxins, hydroxyeicosatetraenoic acids by different LOX enzymes. DHA produces the anti-inflammatory resolvins D (RvD) through the action of LOX enzymes (5-LOX and 15-LOX). The studied PUFA and metabolites in the present research are indicated by blue and red stars, respectively.

**Figure 2 nutrients-11-00966-f002:**
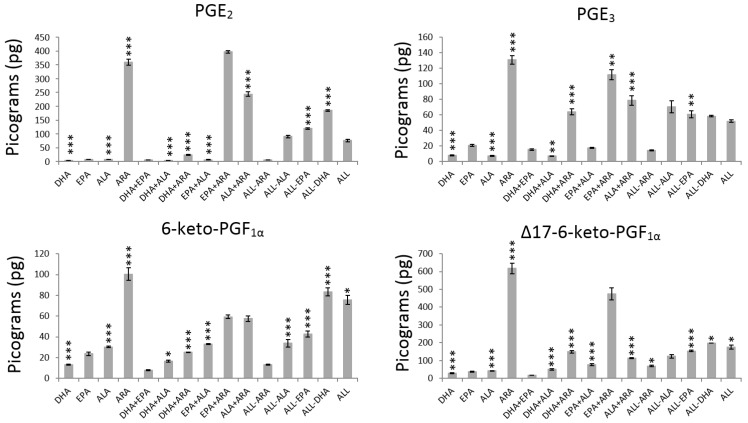
Average production of prostaglandins and prostacyclins by HUVEC exposed to different PUFA combinations. The asterisks indicate significant values at *p* < 0.05 (*), *p* < 0.001 (**) and *p* < 0.000 (***). The actual numerical *p*-values are described in Appendix A.

**Figure 3 nutrients-11-00966-f003:**
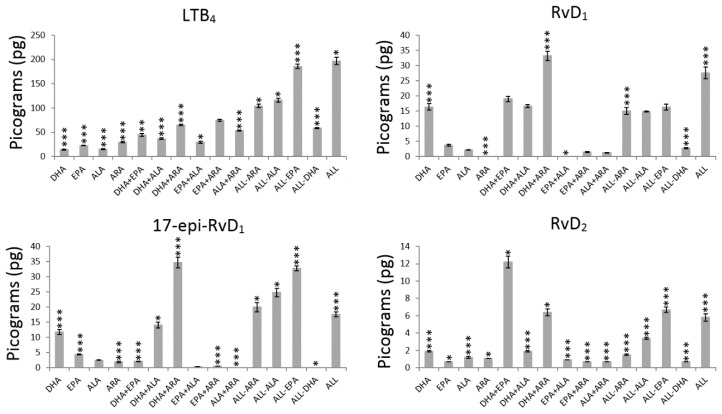
Average production of leukotriene and resolvins by HUVEC exposed to different PUFA combinations. The asterisks indicate significant values at *p* < 0.05 (*), *p* < 0.001 (**) and *p* < 0.000 (***).The actual numerical *p*-values are described in Appendix A.

**Figure 4 nutrients-11-00966-f004:**
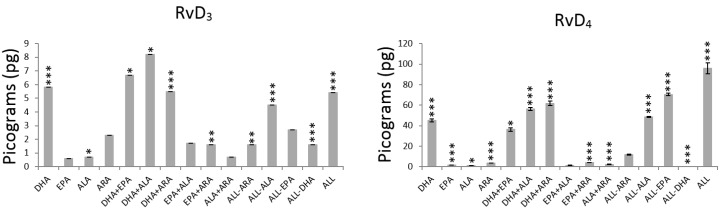
Average production of resolvins by HUVEC exposed to different PUFA combinations. The asterisks indicate significant values at *p* < 0.05 (*), *p* < 0.001 (**) and *p* < 0.000 (***).The actual numerical *p*-values are described in Appendix A.

**Figure 5 nutrients-11-00966-f005:**
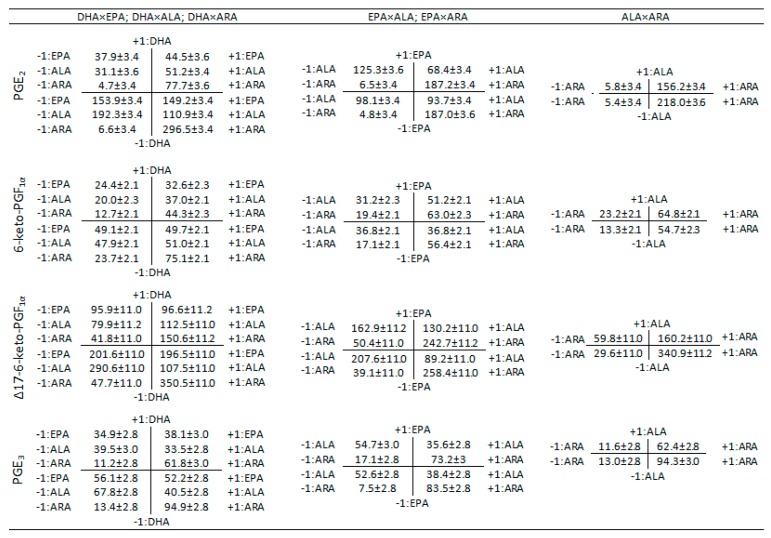
Interaction diagrams for elucidating the effect between two different PUFA on the production of cyclooxygenase metabolites by HUVEC. Explanation about how the diagrams were calculated is given in Section 3.2 and supporting Appendix A. Values in picograms (pg) are expressed as average ± standard error.

**Figure 6 nutrients-11-00966-f006:**
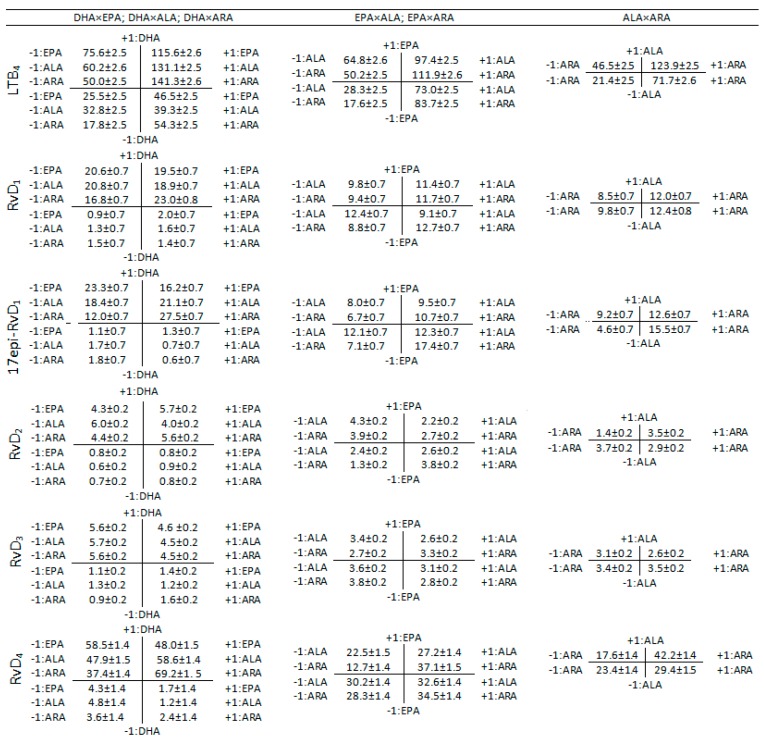
Interaction diagrams for elucidating the effect between two different PUFA on the production of lipoxygenase metabolites by HUVEC. Explanation about how the diagrams were calculated is given in Section 3.2 and supporting Appendix A. Values in picograms (pg) are expressed as average ± standard error.

**Figure 7 nutrients-11-00966-f007:**
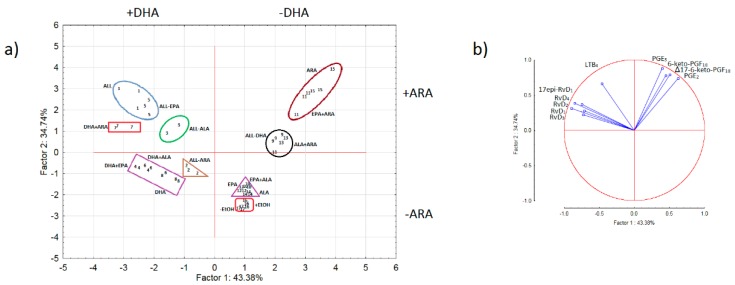
PCA (**a**) score and (**b**) loading plots showing the relationship between the different combinations of ω-3 and ω-6 PUFA according to the proposed 2^4^ factorial design described in Table 1 and the relationship between the different COX and LOX metabolites, respectively. The plots show that DHA and ARA are the main substrates behind the production of LOX and COX metabolites, respectively. DHA is associated with a high production of resolvins, which is counter regulated by ARA through a high production of prostaglandins. The loading plot confirms that 78.12% of the total data variability is explained by the production of resolvins and prostaglandins. The numbers inside the nine different clusters correspond to the different PUFA combinations. 1=ALL; 2=ALL-ARA; 3=ALL-ALA; 4=DHA+EPA; 5=ALL-EPA; 6=DHA+ALA; 7=DHA+ARA; 8=DHA; 9=ALL-DHA; 10=EPA+ALA; 11=EPA+ARA; 12=EPA; 13=ALA+ARA; 14=ALA; 15=ARA; 16=+EtOH; 17=-EtOH.

**Table 1 nutrients-11-00966-t001:** A 2^4^-full factorial design to assess the release of prostaglandins (PGE_2_, PGE_3_), prostacyclins (6-keto-PGF_1α_, Δ17-6-keto-PGF_1α_), leukotriene (LTB_4_) and resolvins (RvD_1_, 17epi-RvD_1_, RvD_2_, RvD, RvD_4_) into endothelial cell growth basal medium-2 (EBM-2) by HUVEC after exposure to the following PUFA: docosahexaenoic acid (DHA), eicosapentaenoic acid (EPA), α-linolenic acid (ALA) and arachidonic acid (ARA). The absence or presence of each fatty acid is designated as −1 and +1, respectively. The amount of metabolite in picograms (pg) is expressed as average ± standard error of three independent culture preparation replicates (*n* = 3). The values were computed by using the individual measurements in Appendix A.

CODE	PUFA					Metabolites								
	ω-3			ω-6	Cyclooxygenase Pathway				Lipoxygenase Pathway					
	DHA	EPA	ALA	ARA	PGE_2_	PGE_3_	6-keto-PGF_1α_	Δ17-6-keto-PGF_1α_	LTB_4_	RvD_1_	17epi-RvD_1_	RvD_2_	RvD_3_	RvD_4_
ALL	+1	+1	+1	+1	76.1 ± 5.0	51.9 ± 1.7	75.6 ± 4.3	175.5 ± 12.4	197.2 ± 7.7	27.6 ± 1.9	17.6 ± 0.8	5.8 ± 0.4	5.4 ± 0.4	95.9 ± 5.4
ALL-ARA	+1	+1	+1	−1	5.7 ± 0.2	14.3 ± 0.7	13.4 ± 0.5	69.7 ± 5.2	104.2 ± 3.5	15.0 ± 1.1	20 ± 1.5	1.5 ± 0.1	1.6 ± 0.1	11.7 ± 0.5
ALL-ALA *	+1	+1	−1	+1	90.4 ± 5.2	70.4 ± 7.8	33.8 ± 3.6	123.5 ± 9.6	116.6 ± 4.0	14.9 ± 0.2	24.8 ± 1.3	3.4 ± 0.1	4.5 ± 0.1	48.6 ± 0.5
DHA+EPA	+1	+1	−1	−1	5.6 ± 0.2	15.6 ± 1.0	7.9 ± 0.5	17.5 ± 1.1	44.8 ± 3.0	19.0 ± 0.9	2.1 ± 0.1	12.2 ± 0.7	6.7 ± 0.1	36.2 ± 1.5
ALL-EPA	+1	−1	+1	+1	119.7 ± 2.5	60.8 ± 4.3	42.7 ± 2.8	154.1 ± 3.4	186.0 ± 4.7	16.3 ± 1	32.7 ± 0.8	6.7 ± 0.3	2.7 ± 0.1	70.5 ± 1.4
DHA+ALA	+1	−1	+1	−1	3.3 ± 0.1	7.0 ± 0.2	16.5 ± 0.9	50.8 ± 4.1	36.9 ± 1.2	16.7 ± 0.5	14.1 ± 1.0	1.9 ± 0.1	8.2 ± 0.6	56.4 ± 1.5
DHA+ARA	+1	−1	−1	+1	24.5 ± 1.4	64 ± 3.7	25.1 ± 0.2	149.4 ± 6.5	65.2 ± 1.9	33.2 ± 1.5	34.7 ± 1.8	6.4 ± 0.4	5.5 ± 0.4	61.7 ± 2.1
DHA	+1	−1	−1	−1	4.2 ± 0.3	7.9 ± 0.6	13.2 ± 0.5	29.2 ± 2.2	14.3 ± 1.1	16.4 ± 1	11.8 ± 0.8	1.9 ± 0.1	5.8 ± 0.5	45.3 ± 1.4
ALL-DHA	−1	+1	+1	+1	184.9 ± 3.0	58.5 ± 0.8	83.3 ± 3.9	198.3 ± 1.0	58.8 ± 0.9	2.8 ± 0.2	0.0 ± 0.0	0.7 ± 0.0	1.6 ± 0.1	0.0 ± 0.0
EPA+ALA	−1	+1	+1	−1	6.8 ± 0.5	17.5 ± 0.6	32.8 ± 0.5	77.1 ± 5.8	29.3 ± 2.0	0.0 ± 0.0	0.3 ± 0.0	0.9 ± 0.0	1.7 ± 0.1	1.3 ± 0.1
EPA+ARA	−1	+1	−1	+1	397.3 ± 4.9	111.9 ± 6.3	59.3 ± 1.7	473.4 ± 33.9	75.1 ± 1.9	1.5 ± 0.1	0.5 ± 0.0	0.7 ± 0.0	1.6 ± 0.1	3.9 ± 0.1
EPA	−1	+1	−1	−1	7.8 ± 0.2	20.8 ± 1.4	23.6 ± 1.5	37.2 ± 2.4	22.8 ± 0.8	3.7 ± 0.3	4.4 ± 0.1	0.7 ± 0.0	0.6 ± 0.0	1.6 ± 0.1
ALA+ARA	−1	−1	+1	+1	244.2 ± 7.6	78.6 ± 6.2	57.6 ± 2.6	113 ± 2.3	53.5 ± 0.7	1.3 ± 0.1	0.0 ± 0.0	0.7 ± 0.0	0.7 ± 0.0	2.3 ± 0.1
ALA	−1	−1	+1	−1	7.5 ± 0.2	7.4 ± 0.5	30.2 ± 0.8	41.7 ± 2.6	15.5 ± 1.2	2.2 ± 0.1	2.5 ± 0.1	1.2 ± 0.1	0.7 ± 0.0	1.2 ± 0.1
ARA	−1	−1	−1	+1	359.7 ± 10.6	130.8 ± 5.4	100.3 ± 6	617.3 ± 28.6	29.9 ± 1.4	0.0 ± 0.0	1.9 ± 0.2	1.1 ± 0.0	2.3 ± 0.2	3.4 ± 0.1
EtOH	−1	−1	−1	−1	4.2 ± 0.3	7.6 ± 0.6	8.4 ± 0.6	34.5 ± 0.5	3.6 ± 0.2	0.0 ± 0.0	0.0 ± 0.0	0.0 ± 0.0	0.5 ± 0.0	10.5 ± 0.6

The term CODE summarizes the various conditions dictated by the factorial design. For example, ALL is the condition where the four PUFAs are added into the medium (the four PUFA at level +1), while ALL-DHA is the condition where EPA, ALA and ARA are at level +1 and DHA was not added (level −1). * Only duplicate preparations were available for ALL-ALA (*n* = 2).

**Table 2 nutrients-11-00966-t002:** Average production of prostaglandins, prostacyclins, leukotriene and resolvins by HUVEC exposed to different PUFA. The values used to compute the averages at level −1 and +1 are those described in Appendix A. Values in picograms (pg) are expressed as average ± standard error.

		Prostaglandins		Prostacyclins		Leukotriene	Resolvins				
PUFA	Level	PGE_2_	PGE_3_	6-keto-PGF_1α_	Δ17-6-keto-PGF_1α_	LTB_4_	RvD_1_	17epi-RvD_1_	RvD_2_	RvD_3_	RvD_4_
DHA	−1	151.6 ± 2.4	54.1 ± 2.0	49.4 ± 1.5	199.1 ± 7.5	36.1 ± 1.7	1.4 ± 0.5	1.2 ± 0.5	0.8 ± 0.2	1.3 ± 0.2	3.0 ± 1.0
	+1	41.2 ± 2.5	36.5 ± 2.0	28.5 ± 1.6	96.2 ± 7.7	95.7 ± 1.8	19.9 ± 0.5	19.7 ± 0.5	5.0 ± 0.2	5.1 ± 0.2	53.3 ± 1.0
EPA	−1	95.9 ± 2.4	45.5 ± 2.0	36.8 ± 1.5	148.8 ± 7.5	50.6 ± 1.7	10.8 ± 0.5	12.2 ± 0.5	2.5 ± 0.2	3.3 ± 0.2	31.4 ± 1.0
	+1	96.8 ± 2.5	45.1 ± 2.0	41.2 ± 1.6	146.6 ± 7.7	81.1 ± 1.8	10.6 ± 0.5	8.7 ± 0.50.5	3.3 ± 0.2	3.0 ± 0.2	24.9 ± 1.0
ALA	−1	111.7 ± 2.5	53.6 ± 2.0	34.0 ± 1.6	185.3 ± 7.7	46.5 ± 1.8	11.1 ± 0.5	10.1 ± 0.5	3.3 ± 0.2	3.5 ± 0.2	26.4 ± 1.0
	+1	81.0 ± 2.4	37.0 ± 2.0	44.0 ± 1.5	110.0 ± 7.5	85.2 ± 1.7	10.2 ± 0.5	10.9 ± 0.5	2.4 ± 0.2	2.9 ± 0.2	29.9 ± 1.0
ARA	−1	5.6 ± 2.4	12.3 ± 2.0	18.2 ± 1.5	44.7 ± 7.5	33.9 ± 1.7	9.1 ± 0.5	6.9 ± 0.5	2.6 ± 0.2	3.3 ± 0.2	20.5 ± 1.0
	+1	187.1 ± 2.5	78.4 ± 2.0	59.7 ± 1.6	250.6 ± 7.7	97.8 ± 1.8	12.2 ± 0.5	14.0 ± 0.5	3.2 ± 0.2	3.1 ± 0.2	35.8 ± 1.0

DHA: docosahexaenoic acid; EPA: eicosapentaenoic acid; ALA: α-linolenic acid; ARA: arachidonic acid. Level −1 without PUFA; Level +1 with PUFA.

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
