# Peer review of "The Effect of Omega-3 and Omega-6 Polyunsaturated Fatty Acids on the Production of Cyclooxygenase and Lipoxygenase Metabolites by Human Umbilical Vein Endothelial Cells"

_nutrients, 2019, doi:10.3390/nu11050966_

Reviewer 1 Report

There are a number of grammatical errors which should be corrected.

The work is very ambitious looking at the effect of fatty acids on a number of end-point metabolites. The Authors are correct in thinking that most papers approach their analyses by looking at the effect on single or few metabolites and either stimulate With a single PUFA or more commonly add a mix (eg EPA and DHA) without trying to understand the interaction between these.

The work presents a lot of data. This is presented both in the results, tables and largely repeated in the discussion. This poses a Challenge for the Reader who is overwhelmed With data. The Authors should look for a better format to structure their work and to present the results and relate these to biological or clinically meaningful conclusions. How does their work contribute to Our knowledge of PUFA actions in the body? What are the next steps in this research or how do these results affect future research?

Author Response

The attached document contains our answers to the report of Reviewer 1

Reviewer 2 Report

In this manuscript, the authors investigated the association between different PUFAs and the release of cyclooxygenase and lipoxygenase metabolites in cell medium by human umbilical vein endothelial cells (HUVEC). The authors report that DHA is the driving force of the beneficial effects of n-3 PUFAs. The manuscript is well written and the data is interesting; however, the following points should be considered to improve the impact of this manuscript.

-       line 112: For how long fatty acids were coupled to FBS and which FBS concentration was used? Which fatty acid to FBS ratio was used (2:1, 3:1, 6:1)?

-       line 115: Did the authors consider having unhealthy vs healthy combinations of n-6 to n-3 fatty acid ratios?

-       line 118: Could the authors comment why they have not measured fatty acid composition which could give another aspect for the interaction between fatty acid metabolism and the cyclooxigenase and lipoxigenase metabolites.

-       line 134 statistics: How the authors have considered correcting for multiple testing?

-       line 154: p values or significance by symbols should be shown in this table2

-       line 176: This section is hard to understand and needs clarification.

-       Table S1 needs clarification. The slash separated values needs explanation. What exactly the values are? This is not clear.

-       Table 3: This table is not clear enough and is hard to understand.

Author Response

The attached document contains our answers to the report of Reviewer 2

Round  2

Reviewer 1 Report

Throughout the work: The authors have added additional text and present the data in new figures which helps to understand the data. The discussion is easier to follow with sub-titling.

Conclusion

Lines 443-446 should be deleted as these are not conclusions but self-made justifications for how data was presented.

I still feel that the conclusions lacks an over-riding picture of what this work means. What is novel, what is pushing our understanding of this area? If the EPA work on PGE3 has previously been published then is their work novel?, or is it supporting a hypothesis that has been shown but not well recognized.

If the work shows that the complexity is greater than previously realized – then what does this mean for future omega-3/6 research, both mechanistic and clinical.  I do not want to say what should be said, but the authors should be bolder and state what they think the relevance of their work is.  The authors have performed a considerable amount of work and this article deserves a better conclusion. This will make it more accessible to Readers.  

Author Response

Throughout the work: The authors have added additional text and present the data in new figures which helps to understand the data. The discussion is easier to follow with sub-titling.

Conclusion

Lines 443-446 should be deleted as these are not conclusions but self-made justifications for how data was presented.

The paragraph in question (lines 443-446) was deleted.

I still feel that the conclusions lacks an over-riding picture of what this work means. What is novel, what is pushing our understanding of this area? If the EPA work on PGE3 has previously been published then is their work novel?, or is it supporting a hypothesis that has been shown but not well recognized.

If the work shows that the complexity is greater than previously realized – then what does this mean for future omega-3/6 research, both mechanistic and clinical.  I do not want to say what should be said, but the authors should be bolder and state what they think the relevance of their work is.  The authors have performed a considerable amount of work and this article deserves a better conclusion. This will make it more accessible to Readers.  

We do agree with the previous comments and in the corrected version, we have changed the conclusion section according to the suggestions of the Reviewer.

Reviewer 2 Report

In line 132, the authors do not report the conditions (which temperature for how long) in which the coupling of the fatty acids to FBS was done. Additionally, in line 182, there's a typo.

Author Response

The temperature and time have been added in the corrected version and the typo mistake was corrected.